# Using an artificial neural network to develop an optimal model of straight punch in boxing and training in punch techniques based on this model and real-time feedback

Ilshat Khasanshin *, Aleksey Osipov

Department of Data Analysis and Machine Learning/Financial University under the Government of the Russian Federation, Moscow, Russia

* iykhasanshin@fa.ru

**Data Availability Statement:** https://github.com/hasanshin/punch_dataset.

**Funding:** The authors received no specific funding for this work.

## Abstract

The work was aimed to develop an optimal model of a straight punch in boxing based on an artificial neural network (ANN) in the form of a multilayer perceptron, as well as to develop a technique for improving the technique of punches in boxing based on feedback, when each punch delivered by a boxer was compared with the optimal model. The architecture of the neural network optimal punch model included an input layer of 600 nodes—the values of absolute accelerations and angular velocities, four hidden ones, as well as a binary output layer (the best and not the best punch). To measure accelerations and angular velocities, inertial measuring devices were attached to the boxers' wrists. Highly qualified participated in the data set for the development of the optimal model. The best punches were chosen according to the criteria of strength and speed. The punch force was determined using a boxing pad with the function of measuring the punch force. In order to be able to compare punches, a unified parameter was developed, called the punch quality, which is equal to the product of the effective force and the punch speed. To study the effects of biofeedback, the boxing pads were equipped with five LEDs. The more LEDs were turned on, the more the punch corresponded to the optimal model. As a result of the study, an almost linear relationship was found between the quality of the punch of entry-level boxers and the optimal model. The use of feedback allowed for an increase in the quality of punches from 11 to 25%, which is on average twice as high as in the group where the feedback method was not used. Studies have shown that it is possible to develop an optimal punch model. According to the degree of compliance with this model, you can evaluate and train boxers in the technique.

**Competing interests:** The authors have declared that no competing interests exist.

## Introduction

Victory in a boxing fight depends on many factors: physical strength and speed, perseverance, will to win, endurance and technical readiness. Technique plays a leading role in boxing, as it is the ability to strike the strongest and fastest in the most optimal way.

Great boxers Joe Frazier, George Foreman, Muhammad Ali, Mike Tyson had the same weight class, but if you consider their technique of performing punches, it is easy to see very big differences. Therefore, the question arises: is there a general ideal, optimal striking technique, or can the striking technique be ideal only when applied to a specific athlete with their neurophysiological characteristics, weight, height, body structure and muscles?

The problem is that currently, most often, the technique of strikes is a scheme under which the coach tries to fit the athlete, which is not always a good strategy. Therefore, it often happens that an athlete has a correct, beautiful technique, but his punches are not very strong and fast, and the athlete has few victories.

Therefore, the work was aimed to develop an optimal model of punch, focusing on which it is possible to improve the technique of punch. It was not just a specific motor pattern that was developed, it was a model that was formed on the basis of dataset from different boxers who differed in weight, strength, height, style and technique of punches. During the experiments, it was necessary to determine whether the degree of compliance with the optimal model correlated with the technique of punches in the control group of boxers who did not participate in the creation of the model. The model was developed on the basis of an artificial neural network in the form of a multilayer perceptron.

Although a machine-trained model is a black box, this model will be very useful for coaching practice and automating scientific research. And the review [1] notes that the application of machine and deep learning to automate data collection in sports has good prospects.

In modern studies [1], video motion capture and IMUs are used to develop the punch model. However, [1] concluded that the advantages of wearable IMUs are that they are affordable, wireless and autonomous in operation. Applications that use computer vision to recognize movements require several pre-processing steps, including detecting and tracking the athlete, clipping time, and recognizing target actions that depend on the sport and type of footage taken. In [2], a review of the use of inertial sensors for analyzing results in martial arts was conducted. The authors analyzed 36 papers in which IMUs was used to determine parameters such as punch quality, classification, frequency, head punch and technique. IMUs was mostly used on the wrists or shins. Data analysis based on machine learning was used in four articles. A review conducted in [3] notes that IMUs are often used in conjunction with visual systems for training movements in martial arts.

Therefore, the following research methodology was developed. During the experiments, each boxer performed a series of punches on a Boxing pad with the function of measuring the punch force. On each of the boxer's hands, the IMU was attached to the wrist, which measured the acceleration and angular velocity of each punch. Thus, for each punch, the punch force, acceleration, and angular velocity were measured. From all the punches, the ones that were characterized by the best indicators of speed and strength were selected. These punches formed a data set for training the artificial neural network. The experimental group included highly qualified athletes with at least 5 years of training experience. These boxers were selected from the group of championship winners at the level not lower than the Russian championship. This group included Champions and prize-winners of international Championships. During the experiments, it was necessary to determine whether the degree of compliance with the optimal model would correlate with the punching technique of the control group of boxers, those who did not participate in the creation of the model.

A similar approach was applied in a very interesting work [4], the purpose of which was to determine the level of readiness for the kinematic parameters of the kick of taekwondo athletes. Seven elite and seven sub-elite athletes were tested for kick-specific variables (KSV, composed of kinematic variables and power of impact) and for concentric isokinetic peak torque (PT) at $60^0/s$ and $240^0/s$. For the analysis, the following method was applied: linear discriminant analysis (LDA). The LDA showed an accuracy of 85.7% (p = 0.003) in predicting expertise level based on hip flexion and extension torques at $240^0/s$ and on knee extension velocity during the kick.

Researchers are also working in this direction [5]. This study aimed to investigate the differences in punching execution between 15 potential Olympic medalist boxers (Elite group) and 8 younger well trained boxers (Junior group). Each athlete was equipped with an instrumented suit composed of 17 inertial measurement units (IMU) and were asked to perform several series of 3 standardized punch types (cross, hook, and uppercut) with maximal force. The results of this study showed that elite boxers systematically produced more force and at a higher speed for the three types of punches compared to juniors. Further analysis revealed differences in joint contribution between Elites and juniors, with juniors presenting a higher shoulder contribution for the three types of strikes. Finally ground reaction force imbalance between the front and rear foot was revealed in the cross only, in all boxers (60.6 ± 24.9 vs. 39.4 ± 24.9% and 54.1 ± 7.1 vs. 45.9 ± 7.1%, p ≤ 0.05, for the front vs. rear foot in Elite and Juniors, respectively) but not different between groups.

The punch force measurement for the development of the optimal punch model was carried out on a punching pad with the punch measurement function. In [6], it is proposed to apply Artificial Intelligence-based Quorum systems for real-time wireless sensor networks. In the article [7], the authors developed a device (a combination of a punching bag with the function of measuring the punch force and IMUs, which is attached to the boxer's wrist) that can determine the boxer's reaction time, the force of punches and the frequency of a series of punches.

In our studies, IMUs were installed on the wrists of athletes, which is justified by many successful studies [8–12]. The authors [8] conducted a study of stroke frequency by installing an accelerometer on the wrist. The study involved 16 people, the acceleration was averaged over the standard deviation, then the calculated acceleration value was used to determine the average integral speed. In this work, the gyroscope, which is usually included in IMUs, was not used. In [9], the authors described a study in which IMUs were used to determine the punches that can be counted in an Amateur Boxing match. The authors [10] applied an integrated approach to the study of movement recognition in fencing—simultaneously IMUs (accelerometer data) and video capture of movements.

The decision to use IMUs only on the wrists of athletes was also based on the intention to develop a system that can then be easily applied in coaching practice.

Many machine learning approaches are used to develop the punch model [1, 2]: CT (Classification Tree), CNN (Convolution Neural Network), KNN (K-Nearest Neighbours), SVM (Support Vector Machine), DTW (Dynamic Time Warping), RNN (Radial Basis Function Neural Network).

To teach the technique of movements in martial arts, the analysis of movements based on F-DTW (Fast Dynamic Time Warping) was used in [13]. In this article, the authors developed a system that recorded athletes' movements using an infrared camera, and classified the data using F-DTW (Fast Dynamic Time Warping), and then issued a report that contained errors in performing the movement and ways to improve it. An accuracy of 91.07% was achieved in the classification of movements.

In work [14], logistic regression, support vector machine, random forest, decision tree, Naive Bayes, K nearest neighbor classifiers are used for prediction, and their accuracy is compared to choose the better machine learning model. SVM provides higher accuracy (96.0) among the chosen algorithms.

In [15], the authors analyzed the level of punch recognition using commercial punch-trackers. Descriptive statistics and multilevel modelling were used to analyze the data. Punch-trackers Corner (CPT), Everlast (EPT), and Hykso (HPT) detected punches more accurately in trained boxers (TR) than untrained punchers, evidenced by a lower percentage error in TR (p = 0.007). The CPT, EPT, and HPT detected straight punches better than uppercuts and hooks, with a lower percentage error for straight punches (p < 0.001). The recognition of punches with CPT and HPT depended on punch order, with earlier punches in a sequence recognized better. In [16], the ANN in the form of a multilayer perceptron was developed for the purpose of automating the data collection of boxers' punches. The input data for ANN was the IMU data that was attached to the boxers' wrist. The accuracy of punch recognition ranged from 87.2 ± 5.4% to 95.33 ± 2.51%.

In paper [12] discusses the development of an automatic classification of boxing punches based on IMUs and machine learning. Several machine learning techniques were used for classification and compared them: IMUs sensors were installed in two ways—in boxing gloves (method 1) and in boxing gloves and on the back (close proximity to the third thoracic vertebrae—method 2). For sensor method 1, a support vector machine (SVM) model with a Gaussian RBF kernel performed the best (accuracy = 0.96), for sensor method 2, a multi-layered perceptron neural network (MLP-NN) model performed the best (accuracy = 0.98).

In our work, we used a well-proven artificial neural network in the form of a multi-layer perceptron.

In addition to the problem of creating an optimal punch model, another problem is that many different models of boxing punches have been developed, based on the analysis of biomechanics, [17–19], however, these models, for all their accuracy, do not help the trainers much in their practical work. Therefore, to test the developed model, a method of teaching the technique of punch based on biofeedback was developed.

The method is based on the fact that a person has a powerful regulatory mechanism, which has long been used in biofeedback techniques and is used, for example, in the rehabilitation of neurological patients, and is often used in sports. And instead of trying to explain to the boxer how to strike correctly, you can give operational information about the quality of punches in order for the athlete's body to find the right technique itself. One of hypotheses of our study was that the mechanisms of self-regulation of the human body allow boxers to unconsciously choose the best technique of punches. The feedback-based boxing technique training method is based on the fact that each punch delivered by a boxer was compared with the best punches in real time and the result of the comparison was reported to the boxer for correction of the punch technique. In work [19], the authors reviewed the use of biofeedback and IMUs for movement training. This article offers an interesting opinion—«wearable technology is leading a revolution in sports». Also, in [19], it is concluded that prediction based on deep learning and IMUs data shows great potential in the development of biomechanical feedback in real time for effective training of human motor skills. It is recommended to combine motion capture and IMUs.

The fact that biofeedback can be effective for training the technique of punching boxers is shown in [11], where it is said that biofeedback via inertial sensors appears to be a potential technique for modifying human movement patterns in both experts and novices. This low-cost technology could be used to support training across sports, rehabilitation and human-computer interactions.

Thus, we can draw conclusions:

1. The development of an optimal model of punch in boxing based on a multilayer perceptron is a little-studied and urgent task.

2. To test the model and study its practical significance, it is necessary to develop a methodology for teaching the technique of punching boxers based on the model.

## Materials and methods

### Design of experiments

Fig 1 shows the general methodology of the experiment.

During the experiments, the boxer punches a boxing pad. Only single (no combinations of punches) straight punches (jab, cross) were studied. The boxing pad, which was attached to the wall, was equipped with a function for measuring the punch force and a Bluetooth module for exchanging data with a computer. On both wrists of the boxer were fixed inertial measuring units (IMUs), which include a gyroscope and an accelerometer, which allow tracking the rotational and translational movements of the hands. On the wrist, along with the IMUs,

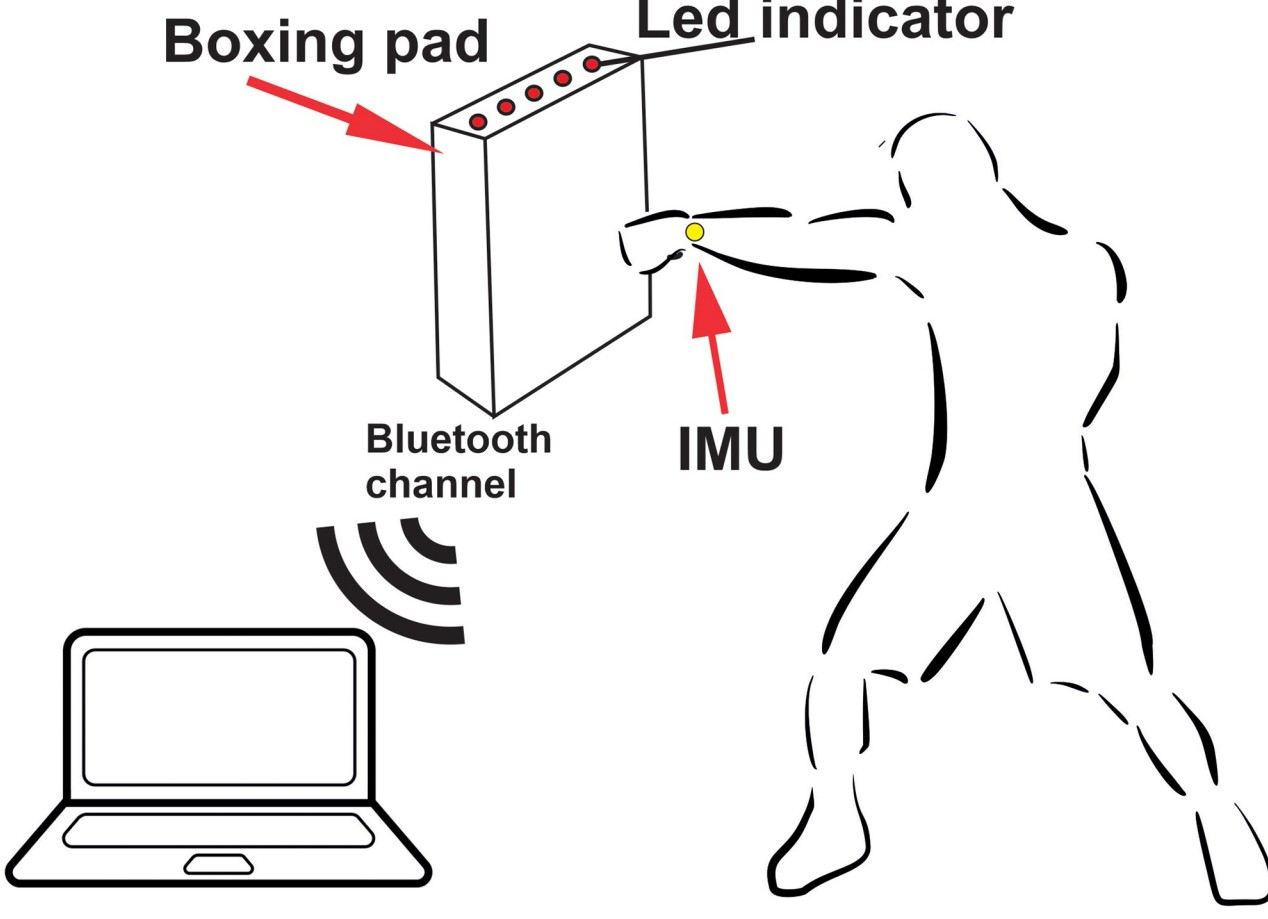

**Fig 1. Design of experiments.**

wireless transmitters were installed, which, via a Bluetooth channel, transmitted data to a computer for analysis using an artificial neural network. Also, five LEDs were installed on the punching pad, which at each stroke signaled the quality of the punch to the boxer—the better the punch (stronger, faster), the more LEDs turn on and give feedback to the boxer. The boxers wore standard sportswear and 10-ounce boxing gloves.

## Measurement system

The measuring system consisted of a punching pad with the function of measuring the punch force and a led indication of the punch quality and inertial measurement units (IMUs) that were attached to the boxers' wrists. All measurement modules were combined together with the computer into a unified system using a Bluetooth channel. The photo and schematic diagram of the punching pad are shown in Fig 2.

The principles of operation of the punching pad are shown in Fig 2. Inside the punch-ing pad 1 there is a air camera 2. When the air camera is punch, the pressure in it increases in proportion to the force of the punch. The pressure is measured using an air pressure sensor 3. The pressure sensor data is converted into an analog-to-digital converter of the microcontroller, which is installed in the control unit 4. The Microcontroller transmits information about the pressure in the air camera via Bluetooth-channel to the computer. Also, information about the punch quality is sent wirelessly from the computer. Based on this data, the microcontroller controls LEDs 5, the bet-ter the punch quality, the more LEDs turn on. The IMU along with the microcontroller and the Bluetooth module were installed in a box weighing 35 grams. This box was attached to the boxer's wrist with a boxing hand wrap (Fig 3).

The IMUs was set so that the positive Y axis of the accelerometer was directed along the direction of punch, and the X axis was directed towards the thumb (Fig 3). The angular

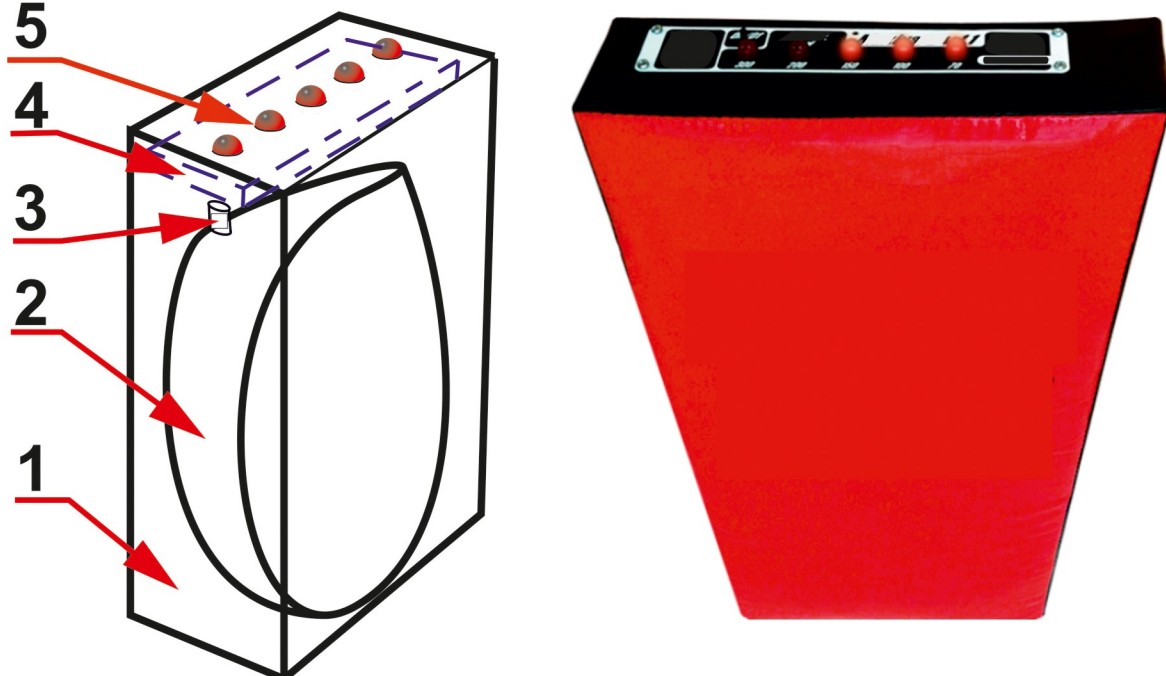

**Fig 2. Photo and schematic diagram of a punching pad.** 1—housing, 2—air camera, 3—air pressure sensor, 4—microcontroller control unit with Bluetooth module, 5—LEDs.

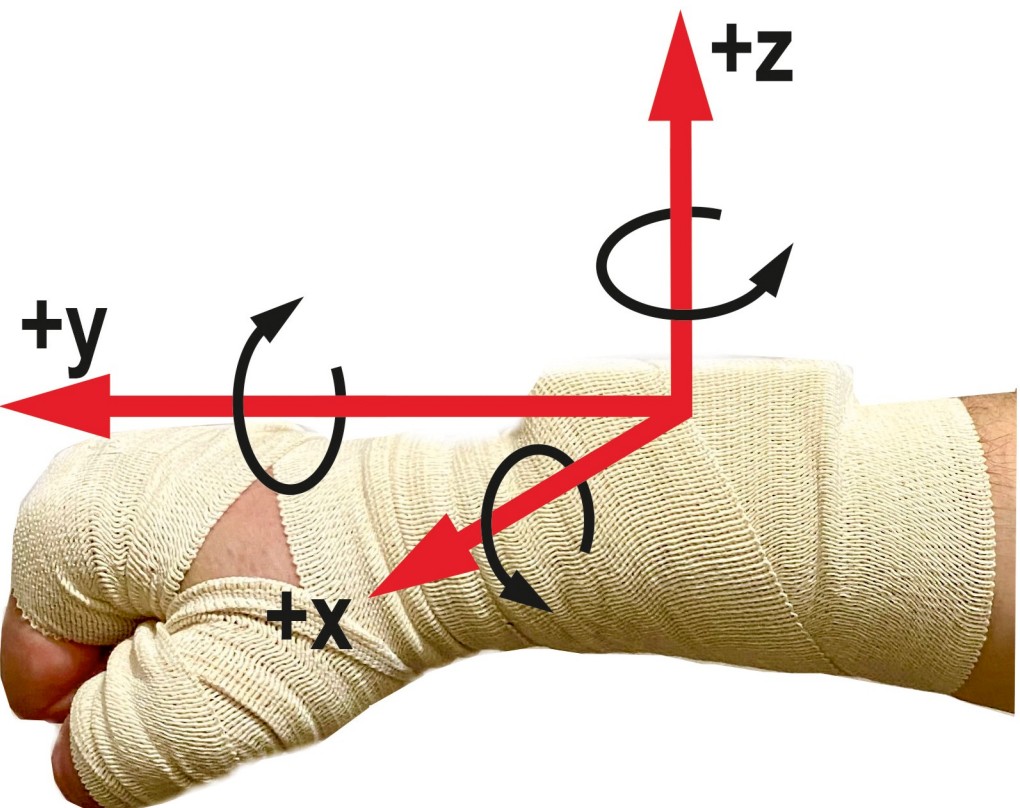

**Fig 3. Measuring module attached to the arm with a boxing hand wrap.** The positive directions of the accelerometer axes are shown, as well as the positive directions of angular rate.

velocity directions are also shown in (Fig 3). The box with the installed IMU, microcontroller was small, so it fit comfortably in a boxing glove (Fig 4).

The IMUs were digital, they are controlled and transmit data via a digital protocol (in our case, the I2C interface was used).

Technical characteristics of accelerometer: sensitivity = ± 16g, non-linearity = 0.2%. Gyroscope: sensitivity = ± 2000˚ / sec, nonlinearity = 0.5%.

The measurement modules were connected to microcontroller devices. Since accelerometers and gyroscopes are characterized by high data noise, the microcontroller processed the accelerometer and gyroscope data using a Kalman filter. The microcontroller also controlled data transfer from the IMU to the computer using the Bluetooth module.

## Neural network architecture

For the purposes of analysis and classification of boxing punches, a neural network model in the form of a multilayer perceptron was used [20]. The neural network architecture was as follows: input layer—600 nodes (values of absolute accelerations and angular rates); hidden layers –512 nodes -> 256 nodes -> 128 nodes -> 64 nodes; output layer—2 nodes (best punch, not best punch).

Data processing was carried out using the Keras library. Keras is a deep learning API written in Python, running on top of the machine-learning platform TensorFlow. It was developed with a focus on enabling fast experimentation [21]. The activation function determines the

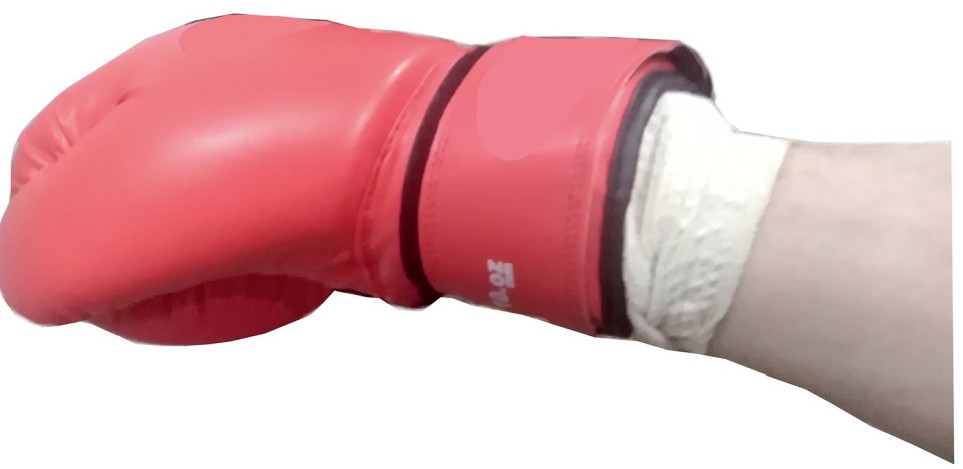

**Fig 4. Boxing glove with measuring module.**

output value of the neuron depending on the result of the weighted sum of the inputs and the threshold value. In our work, the sigmoid activation function was applied on each layer.

When training an artificial neural network, the loss function was minimized, which, when using the Keras library, was specified as a parameter of the compile method (sets up the model for training) of the Model class [21]. Since the data in our case has only 2 categories, the loss function in the form of binary cross-entropy was used. The Adam optimization algorithm was used for the model. Adam optimization is a stochastic gradient descent method that is based on adaptive estimation of first-order and second-order moments [21]. The configuration of optimization using the Adam algorithm in the Keras library [23] was applied as follows:

$$keras.optimizers.Adam(learning\_rate = 0.001, beta\_1 = 0.9, beta\_2 = 0.999), \quad (1)$$

where beta_1, beta_2 is the exponential decay rate for the 1st and 2nd order moment estimates, respectively.

To select the optimal model and prevent overfitting, the k-Fold method of cross-validation was chosen. This is a data resampling procedure for training and model validation that is used to evaluate the ANN models on a limited data sample. The sklearn library was used for cross-validation [22]. The sklearn library was used for cross-validation [23]. The k-Fold cross-validation procedure consisted of the following steps:

1. The received data for training (dataset) shuffled randomly.

2. The dataset is divided into k sample groups (in this research, k = 10)

3. For each unique sample:

   - installed as a testing dataset;

   - the remaining groups are used to train the model;

   - the model is evaluated on a test sample;

   - the model score is saved, and the next selection is moved on.

4. The model quality parameters are generalized based on the model estimates obtained from the samples.

Classification estimates for various factors were made on the basis of the F-score. This metric was also carried out on the basis of the skearn library [24]. The F-score for binary classification has the form F1-score:

$$F1 = \frac{2(PR * RC)}{PR + RC} \qquad (2)$$

where PR is the precision of the model (3), RC is the recall or sensitivity of the model (4).

Precision is the percentage of correct answers of the model:

$$PR = \frac{TP}{TP + FP} \qquad (3)$$

where TP is the number of cases correctly assigned to the "best punch", FP is the number of cases where the cases were assigned to the "best punch", but were not.

Recall determines the number of defined true positive cases, i.e., those assigned to "best punch", among all class labels that have been identified as "best punch":

$$RC = \frac{TP}{TP + FN} \qquad (4)$$

where TP is the number of cases correctly assigned to the "best punch", FN is the number of cases where the cases were not assigned to the "best punch", but were.

## Experiment

Three groups of boxers of the Boxing Federation of the Republic of Tatarstan (Kazan, Russia) participated in the experiments. The study was conducted in accordance with the Declaration of Helsinki, and the protocol was approved by the Ethics Committee of Financial University under the Government of the Russian Federation № 20210902b. Group 1: gender—male; number—10 people; age—from 23 to 32 years old; weight—3 people up to 57 kg (featherweight), 4 people up to 72 kg (Middleweight), 2 people up to 91 kg (cruiserweight), 1 person— 96 kg (heavyweight); experience—over 5 years of training; achievements—3 athletes are members of the Russian national Boxing team, all athletes are Champions and prize-winners of the Republic of Ta-tarstan and Russia Boxing Championships.

Group 2: gender—male; number—10 people; age-from 18 to 23 years; weight—4 people up to 57 kg (featherweight), 5 people up to 72 kg (average weight), 1 person—91 kg (cruiserweight); experience: up to 2 years of training. Group 3: gender—male; number—10 people; age-from 18 to 22 years; weight—5 people up to 57 kg (featherweight), 3 people up to 72 kg (average weight), 1 person—90 kg (cruiserweight); experience: up to 2 years of training.

The experiment consisted of three series. In the first series, datasets were collected to create an optimal punch model. This series of experiments involved first group of experienced boxers, whose task was to do only single, no combined punches. Each athlete hit the punching pad 500 times with both hands with a straight punch. The experiment was limited to the study of straight punch techniques. To form the cate-gory of best punches, 5% of the best punches were selected from all experimental data, which were selected according to the criteria of force and velocity. Also, each boxer punched 300 times with each hand in order to collect a test dataset.

The force of the punch was recorded using pressure sensors that measured the pressure in the air camera (Fig 3). However, the force of punch, expressed in Newtons, does not correctly reflect the effect of the punch. If a person of great weight and great physical strength makes a push, then this will not be equivalent to a punch, which the same person with the same force will do in 50 milliseconds, that is, in fact, you need to take into account the power of the

punch:

$$P = \frac{w}{t} \tag{5}$$

where P[W] is power, W[J] is work, and t[s] is time.

Mechanical work is known to be directly proportional to force and distance. In this work, we do not need exact power values, it is enough to make estimates. Therefore, we determine the value, which we call the effective punch force:

$$F_{eff} = \frac{F}{t} \tag{6}$$

where $F_{eff}$ is effective punch force, F is force of punch, and t is time of punch.

Now value $F_{eff}$ will determine, the larger the punch and the shorter the punch time, the more effective the punch. Time in our study is the time of punch from the beginning of the increase of force to the maximum. The second value that determines the effectiveness of the punch is the velocity. The speed changes during the strike in a complex way, but for the best punch, it matters how fast the fist hits the target. This moment was recorded by a signal from the punching pad. This is the moment when the pressure in the air camera has changed to a certain value.

In the second series of experiments, it was determined whether the best punches of group 2 boxers would most closely match the optimal model of punches that was developed in the first series of experiments. And whether there is a relationship between the degree of convergence of the model and the punch and the degree of punch efficiency. In this series, each boxer from this group made 500 right and left punches.

It was assumed that the third series of experiments should be carried out only if the second series of experiments was successful. The third series of experiments involved the second and third groups of boxers. Initially, the effectiveness of each boxer's punches was set for each boxer. Then, the second group was trained using the feedback technique, that is, with each punch on the punching pad, LEDs were turned on. The number of LEDs corresponded to the punch efficiency. Five LEDs turned on meant the most effective punch, that is, the most consistent with the optimal punch model that was developed in the first series of experiments, one led turned on meant the least effective punch. The third group trained as usual, without using feedback. The third series of experiments lasted 1 month.

## Results and discussion

In the first series, we first had to determine 5% of the best punches of 1 group of experienced boxers to collect a dataset of the best punches model. Tables 1 and 2 shows the lower limits of the best punches of all boxers. Table 1 shows left-handed punches, and Table 2 shows right-handed punches. $F_{eff}$ was calculated by the Eq (6).

The average value of the series and the confidence interval 0.95 were calculated. Figs 5 and 6 show the results of a series of punches made by 10 boxers with the left and right hands, according to the effectiveness and speed of the punch, respectively.

Also, to evaluate the punch, a quantity was invented that we called "punch quality" ($q_p$ in Tables 1 and 2), the product of the effective force and speed of the punch. The obtained experimental data can be compared with the data of [7]. In [7], there is no indication of what type of punch they studied, but from the design of the experimental setup, it can be concluded that this is a straight punch. The punch velocity is comparable to the data obtained in [8].

Thus, two categories of punches were selected—"best punch" and "not-best punch", after which the model was trained. During the training of an artificial neural network, an iterative

**Table 1. Left-handed punches.**

| Boxer | Force, H | t,s | $F_{eff}$ | Velocity, m/s | $q_p$ |
|---|---|---|---|---|---|
| 1 | 884.16 | 0.169 | 5231.72 | 12.1 | 63303.76 |
| 2 | 821.82 | 0.141 | 5828.51 | 13.01 | 75828.92 |
| 3 | 921.81 | 0.171 | 5390.70 | 9.84 | 53044.51 |
| 4 | 790.22 | 0.21 | 3762.95 | 11.54 | 43424.47 |
| 5 | 906.56 | 0.154 | 5886.75 | 9.91 | 58337.72 |
| 6 | 994.77 | 0.177 | 5620.17 | 10.4 | 58449.76 |
| 7 | 963.12 | 0.192 | 5016.25 | 9.21 | 46199.66 |
| 8 | 1096.96 | 0.181 | 6060.55 | 9.11 | 55211.63 |
| 9 | 1285.42 | 0.201 | 6395.12 | 7.47 | 47771.58 |
| 10 | 1395.89 | 0.231 | 6042.81 | 6.98 | 42178.84 |

optimization called gradient descent occurs. Each training cycle is usually called epochs. The training process for left-handed punches can be seen in Fig 7, this is a graph of the change in the accuracy of the model and the loss function. For right-handed punches, this process is shown in Fig 8. From the graphs in Figs 5 and 6, it is easy to see that the accuracy of the model became equal to 1 already by the 30-40 epoch. At the same time, the evaluation of the model on test data ("Test" graphs) showed good validation of the model.

To select the optimal model, k-Fold cross-validation was applied, the results of which are shown in Table 3.

The second series was to determine whether an optimal hand punch model could be created or whether this model could only be developed for each boxer individually. Also, as in the first series of experiments, the values of the effective force, velocity and quality of the punches were determined. In order to be able to compare the results of different boxers, the data of their quality of punches were normalized to the maximum value. That is, the relative quality of punches was calculated:

$$qr_p = \frac{q_p}{q_{pmax}} \tag{7}$$

where $q_p$ is the value of the punch quality, $q_{pmax}$ is the maximum value of the punch quality in the series for this boxer, $q_{rp}$ is the relative quality of the punch. Statistical analysis was performed in Statistica, a software package for statistical analysis developed by StatSoft, which

**Table 2. Right-handed punches.**

| Boxer | Force, H | t,s | $F_{eff}$ | Velocity, m/s | $q_p$ |
|---|---|---|---|---|---|
| 1 | 924.61 | 0.155 | 5965.23 | 13.23 | 78919.94 |
| 2 | 921.97 | 0.138 | 6680.94 | 14.32 | 95671.09 |
| 3 | 928.43 | 0.164 | 5661.16 | 10.02 | 56724.81 |
| 4 | 827.99 | 0.234 | 3538.42 | 10.86 | 38427.23 |
| 5 | 936.11 | 0.16 | 5850.69 | 9.08 | 53124.24 |
| 6 | 1100.56 | 0.157 | 7009.94 | 11.15 | 78160.79 |
| 7 | 985.82 | 0.186 | 5300.11 | 9.16 | 48548.98 |
| 8 | 1121.23 | 0.179 | 6263.85 | 10.85 | 67962.82 |
| 9 | 1295.42 | 0.984 | 1316.48 | 7.55 | 9939.452 |
| 10 | 1402.26 | 0.2 | 7011.30 | 6.03 | 42278.14 |

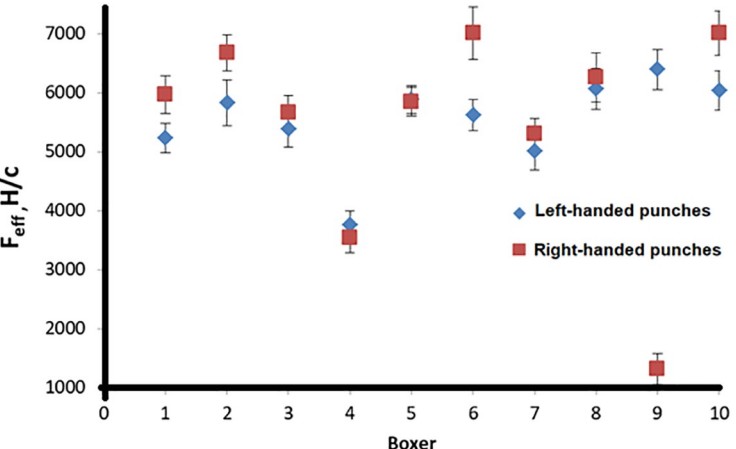

**Fig 5. The effectiveness of 10 boxers' punches made with the left (rhombus) and right (square) hands.**

implements the functions of data analysis, data management, data mining, data visualization using statistical methods.

Relative punch quality correlated to degree of coincidence of the punch quality and the model value with r values of 0.85, respectively (p<0.05).

Fig 9 shows a graph of the relative punch quality versus accuracy. Accuracy means in this case how close the punch value is to the model quality that was obtained in the first series of punches.

Based on the software package Statistica, a polynomial interpolation of the data set was performed, by which a polynomial of the lowest possible degree that passes through the points of the data set was found. Interpolation equation is close to linear:

$$qr_p = 0.31 + 0.32x + 0.29x^2 \qquad (8)$$

where $q_{rp}$—relative punch quality, x—degree of coincidence of the punch quality and the model value.

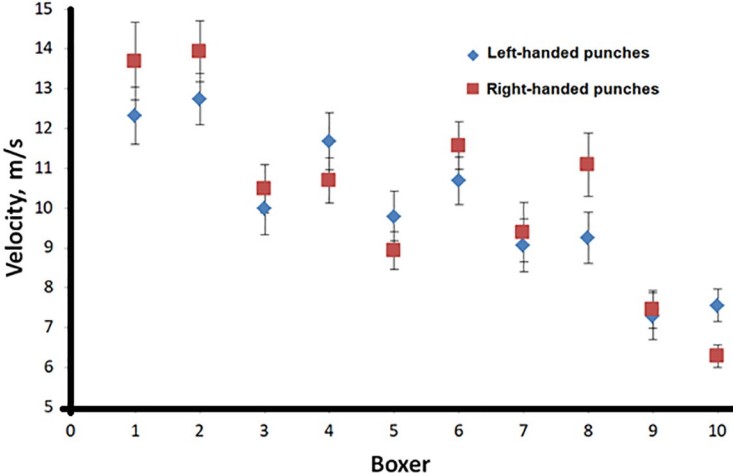

**Fig 6. The effectiveness of 10 boxers' punches made with the left (rhombus) and right (square) hands.**

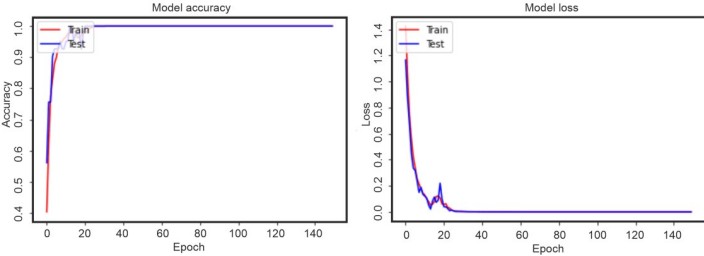

**Fig 7. Accuracy of the model and the loss function for left-handed punches.**

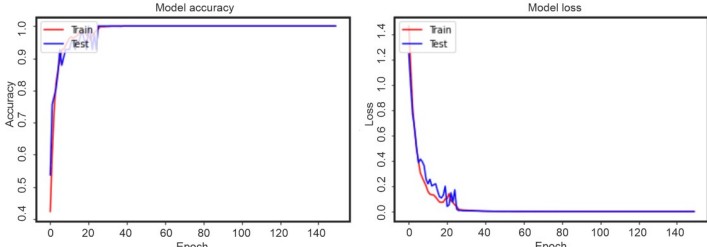

**Fig 8. Accuracy of the model and the loss function for right-handed punches.**

It can be noticed that in the region of low punch quality (up to 0.3), the convergence with the model is poor and a large spread of values can also be observed. This can be explained by the fact that with poor punch quality, there does not seem to be a unified punch model.

Thus, we took the field of linear and angular velocities of the punch, compared it with the model and found that the closer their coincidence, the better the punch.

In [24] where the authors developed a virtual training system for Chinese gymnastics, was not used detection of movement, and compared the degree of compliance with the provisions of the main segments of the body model. The authors [24] obtained results that allowed us to develop a system of teaching the technique of Chinese gymnastics. In [13], we also tried to obtain a model of karate movements based on Kinect, and then compare it with this model of movement for training purposes. The best results in comparing the data were obtained using

**Table 3. Results of k-Fold cross-validation.**

| Sample | Losses of model | Accuracy of model, % |
|---|---|---|
| 1 | 0.7920 | 60 |
| 2 | 1.3998 | 80 |
| 3 | 1.2608 | 80 |
| 4 | 1.2929 | 80 |
| 5 | 0.0594 | 100 |
| 6 | 0.0034 | 100 |
| 7 | 0.0024 | 100 |
| 8 | 0.0020 | 100 |
| 9 | 0.0031 | 100 |
| 10 | 0.0046 | 100 |

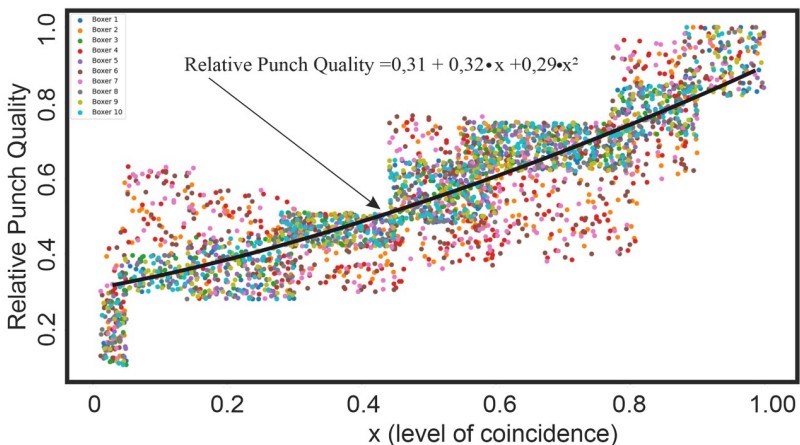

**Fig 9. Dependence of the relative punch quality on the degree of coincidence of the punch quality and the model value.**

F-DTW. The coincidence of the best movements with the model ones was obtained at the level of 91.07%.

After the hypothesis that it was possible to create a better punch model was confirmed, a third series of experiments was conducted, in which training was conducted on the basis of the resulting model.

The experiment involved group 2 and group 3. Both groups were engaged in the same programs but in different gyms. During training on the technique of punches, the second group of boxers worked on Boxing pad, which had five LEDs. The number of LEDs turned on was determined by the quality of the punch, i.e., how much the punch was consistent with the best punch model developed in the first series of experiments.

Fig 10 shows a graph of the change in the quality of the punch for each boxer of group 2 after a month of training. In the second group, there was an increase in the quality of punches from 11 to 25%, with an average increase of 17.51%.

Fig 11 shows a graph of the change in the quality of the punch for each boxer of group 3 after a month of training.

In the group 3, there was an increase in the quality of punches from 5.94 to 12%, with an average increase of 8.56%.

Thus, the feedback technique showed high efficiency. This is in good agreement with the work [4], which describes the use of biofeedback in sports shooting. Feed-back in the work [4]

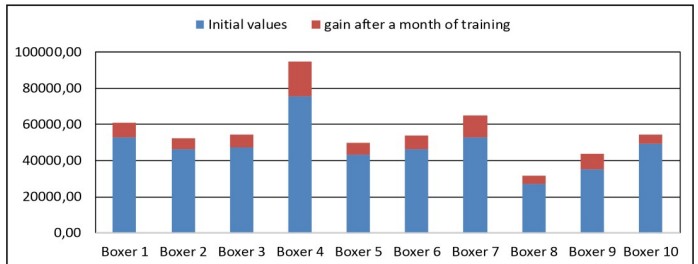

**Fig 10. A graph of the punch quality, group 2.**

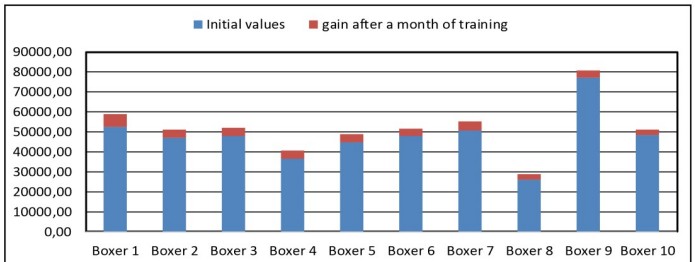

**Fig 11. A graph of the punch quality, group 3.**

was made using error recognition based on machine learning and led to an increase in shooting accuracy and a shorter training period.

In a review conducted in [25], the goal was to determine whether wearable feedback devices improve running performance in runners. The review showed that the positive peak acceleration associated with a stress fracture of the tibia was significantly reduced when receiving its biofeedback. By providing biofeedback to increase step frequency, participants had a significant reduction in vertical load on the knee and ankle and a significant increase in knee flexion. That is, the runners' running technique improved significantly. In the work [11], a study was carried out of the influence of biofeedback on the performance and technique of boxing jab. Sixteen participants (8 novices and 8 experts) performed boxing jabs against the bag in blocked phases of biofeedback. When compared to baseline, the acute effects of externally focused biofeedback on peak bag acceleration were possibly positive in both retention phases for novices ($d = 0.29$; $d = 0.41$) and likely positive for experts ($d = 0.41$; $d = 0.30$), respectively. In [11], the feedback was carried out using an audio signal, the volume of which correlated with the peak value of the acceleration of the boxing bag. That is, in this work there was no comparison with the model of the punch technique.

## Conclusion

On the basis of an artificial neural network in the form of a multilayer perceptron, a straight punch model was developed. High-level boxers participated in collecting the dataset and data for model validation. The criteria for the best punch were chosen velocity and force of punch. The velocity of punch was measured using IMUs, which were attached to the hands of the boxers. The punch force was measured using a punching pad with the force measurement function. The input parameters of the artificial neural network were linear and angular velocities of the boxer's fist. Then experiments were conducted with a group of entry-level boxers of similar age and weight to the first group. In this experiment, we tested the hypothesis that the better the punch technique, the closer its parameters are to the ideal, model ones. In order to be able to compare punches, a unified parameter was developed, called the quality of the punch, which is equal to the product of the effective force and velocity of the punch. The effective punch force is a characteristic proportional to the punch power. As a result of the experiment, a relationship was found between the quality of the punch and the degree of compliance with the model. This dependence turned out to be quite close to linear. Thus, we can conclude that there is an optimal punch model, which is obtained on the basis of the field of linear and angular velocities and which can be used as the best punch technique.

A series of experiments was also conducted to determine the practical significance of the resulting model. For this purpose, two groups of approximately the same level of training, weight and age spent a month training on a similar program. In one of the groups, training

was carried out on a special punching pad, which was equipped with five LEDs. These LEDs were turned on at punch and the closer the punch was to the model, the more LEDs were turned on, that is, so real-time feedback was organized. The second group trained according to the usual program. As a result, after a month of training, the first group had an average of more than twice as good punch quality as the second group. In addition, the increase in the quality of strikes in the first group occurred with impressive dynamics. Thus, this indirectly confirms the hypothesis that it is possible to develop an optimal model of a boxing punch, and also shows the effectiveness of using this model together with real-time feedback.

In the studies, raw IMUs data was used as a dataset, and in further studies, it is planned to apply pre-processing of the data (normalization, reduction of the dimension of the input data, etc.).

Only a straight boxing punch was chosen for the study, as it is the simplest in terms of coordination characteristics. However, based on the study, it can be assumed that for other boxing punches (hook, swing, uppercut), it will also be possible to develop an optimal model. Therefore, it is planned to conduct research in order to develop an optimal model of other boxing punches.

## Acknowledgments

The author would like to thank athletes, coaches and the leadership of the Boxing Federation of the Republic of Tatarstan (Kazan, Russia) for organizing the experiments and actively participating in the study.

## Author Contributions

**Conceptualization:** Ilshat Khasanshin.

**Data curation:** Aleksey Osipov.

**Formal analysis:** Aleksey Osipov.

**Investigation:** Ilshat Khasanshin.

**Methodology:** Ilshat Khasanshin.

**Software:** Ilshat Khasanshin.

**Validation:** Aleksey Osipov.

**Visualization:** Ilshat Khasanshin.

**Writing – original draft:** Ilshat Khasanshin.

**Writing – review & editing:** Ilshat Khasanshin.

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
