## [Decision Letter · Decision Letter 0]

2 Jun 2021

PONE-D-21-15481

Using an artificial neural network to develop an optimal model of straight punch in boxing and training in punch techniques based on this model and real-time feedback

PLOS ONE

Dear Dr. Khasanshin,

Thank you for submitting your manuscript to PLOS ONE. After careful consideration, we feel that it has merit but does not fully meet PLOS ONE’s publication criteria as it currently stands. Therefore, we invite you to submit a revised version of the manuscript that addresses the points raised during the review process.

As per the comments received from the reviewers and my own observation, I recommend major revisions for the paper.

We look forward to receiving your revised manuscript.

Kind regards,

Thippa Reddy Gadekallu

Academic Editor

PLOS ONE

Journal Requirements:

Please provide additional details regarding participant consent. In the ethics statement in the Methods and online submission information, please ensure that you have specified (1) whether consent was informed and (2) what type you obtained (for instance, written or verbal, and if verbal, how it was documented and witnessed). If your study included minors, state whether you obtained consent from parents or guardians. If the need for consent was waived by the ethics committee, please include this information.

We note that you have stated that you will provide repository information for your data at acceptance. Should your manuscript be accepted for publication, we will hold it until you provide the relevant accession numbers or DOIs necessary to access your data. If you wish to make changes to your Data Availability statement, please describe these changes in your cover letter and we will update your Data Availability statement to reflect the information you provide.

Thank you for stating the following financial disclosure:

No.

Thank you for stating the following in your Competing Interests section: 

NO

Reviewers' comments:

Reviewer's Responses to Questions

**Comments to the Author**

1. Is the manuscript technically sound, and do the data support the conclusions?

Reviewer #1: Yes

Reviewer #2: Yes

2. Has the statistical analysis been performed appropriately and rigorously? 

Reviewer #1: Yes

Reviewer #2: Yes

3. Have the authors made all data underlying the findings in their manuscript fully available?

Reviewer #1: Yes

Reviewer #2: No

4. Is the manuscript presented in an intelligible fashion and written in standard English?

Reviewer #1: Yes

Reviewer #2: No

5. Review Comments to the Author

Reviewer #1: 1. Introduction section can be extended to add the issues in the context of the existing work

2. Literature review techniques have to be strengthened by including the issues in the current system and how the author proposes to overcome the same.

3. What is the motivation of the proposed work?

4. Research gaps, objectives of the proposed work should be clearly justified.

5. The authors should consider more recent research done in the field of their study (especially in the years 2018 and 2020 onwards). 6. The paper needs to provide significant experimental details to correctly assess its contribution: What is the validation procedure used?

7. Kindly provide several references to substantiate the claim made in the abstract (that is, provide references to other groups who do or have done research in this area).

8. An error and statistical analysis of data should be performed.

9. The conclusion should state scope for future work.

10. Discuss the future plans with respect to the research state of progress and its limitations.

11. Kindly refer the below paper:

1. Rajput, D.S., Basha, S.M., Xin, Q. et al. Providing diagnosis on diabetes using cloud computing environment to the people living in rural areas of India. J Ambient Intell Human Comput (2021). https://doi.org/10.1007/s12652-021-03154-4

Reviewer #2: This paper has potential but few things need to be arranged

Introduction lacks contribution and structure is missing

Figures need to be improved

You need to top up the literature review

suggested should be applied https://ieeexplore.ieee.org/abstract/document/9430519 and Data collection method https://www.sciencedirect.com/science/article/abs/pii/S0140366420318442

Conclusion should be improved

grammar should be improved

6. PLOS authors have the option to publish the peer review history of their article (what does this mean?). If published, this will include your full peer review and any attached files.

Reviewer #1: No

Reviewer #2: No

---

## [Author Response · Author response to Decision Letter 0]

29 Jun 2021

In general, we would like to thank the reviewers, whose comments helped us to take a fresh look at our work and significantly improve it.

Reviewer #1:

1. Introduction section can be extended to add the issues in the context of the existing work. 

2. Literature review techniques have to be strengthened by including the issues in the current system and how the author proposes to overcome the same.

4. Research gaps, objectives of the proposed work should be clearly justified.

These are interrelated questions and comments. We agree with all the comments, so we have completely redone the structure and content of the introduction. Each hypothesis, goal, and choice of research methods is justified by the work of other authors in this field.

3. What is the motivation of the proposed work?

Sports science is actively developing, while training in the technique of punches in boxing is still intuitive. The student imitates the movements of the teacher and this process does not yet have measurable parameters. Research on the biomechanics of punches almost does not help coaches, you need a model that fits well into coaching practice.

5. The authors should consider more recent research done in the field of their study (especially in the years 2018 and 2020 onwards).

We have extended the list of peer-reviewed papers in this area, especially with articles published over the past year.

6. The paper needs to provide significant experimental details to correctly assess its contribution: What is the validation procedure used?

We have supplemented the article K-fold method of cross-validation of the model. We also supplemented the article with statistical data verification.

7. Kindly provide several references to substantiate the claim made in the abstract (that is, provide references to other groups who do or have done research in this area).

The abstract has been significantly revised and expanded, and it is limited to 300 words, so references to similar works are included in the sections "Introduction" and " Results and discussion". 

8. An error and statistical analysis of data should be performed.

We supplemented the article with statistical data verification. Statistical analysis was performed in Statistica, a software package for statistical analysis developed by StatSoft, which implements the functions of data analysis, data management, data mining, data visualization using statistical methods. 

The average value of the series punches and the confidence interval 0.95 were calculated. 

9. The conclusion should state scope for future work.

10. Discuss the future plans with respect to the research state of progress and its limitations.

Very good point. In the studies, raw IMUs data was used as a dataset, and in further studies, it is planned to apply pre-processing of the data (normalization, reduction of the dimension of the input data, etc.).

Only a straight boxing punch was chosen for the study, as it is the simplest in terms of coordination characteristics. However, based on the study, it can be assumed that for other boxing punches (hook, swing, uppercut), it will also be possible to develop an optimal model. Therefore, it is planned to conduct research in order to develop an optimal model of other boxing punches.

11. Kindly refer the below paper:

1. Rajput, D.S., Basha, S.M., Xin, Q. et al. Providing diagnosis on diabetes using cloud computing environment to the people living in rural areas of India. J Ambient Intell Human Comput (2021). https://doi.org/10.1007/s12652-021-03154-4

A useful article, we have included it in the work.

Reviewer #2: 

Introduction lacks contribution and structure is missing

We have completely redone the structure and content of the introduction. Each hypothesis, goal, and choice of research methods is justified by the work of other authors in this field.

Figures need to be improved

All figures have a resolution of 600 dpi, in accordance with the requirements of PLOS ONE.

You need to top up the literature review

We have extended the list of peer-reviewed papers in this area, especially with articles published over the past year.

suggested should be applied https://ieeexplore.ieee.org/abstract/document/9430519 and Data collection method https://www.sciencedirect.com/science/article/abs/pii/S0140366420318442

A useful article, we have included it in the work.

Conclusion should be improved

We have reworked the conclusion, including our future research plans in this area.

grammar should be improved

We tried to improve the grammar of the article

---

## [Decision Letter · Decision Letter 1]

20 Oct 2021

Using an artificial neural network to develop an optimal model of straight punch in boxing and training in punch techniques based on this model and real-time feedback

PONE-D-21-15481R1

Dear Dr. Khasanshin,

We’re pleased to inform you that your manuscript has been judged scientifically suitable for publication and will be formally accepted for publication once it meets all outstanding technical requirements.

Kind regards,

Bijan Najafi

Academic Editor

PLOS ONE

Additional Editor Comments (optional):

Reviewers' comments:

Reviewer's Responses to Questions

**Comments to the Author**

1. If the authors have adequately addressed your comments raised in a previous round of review and you feel that this manuscript is now acceptable for publication, you may indicate that here to bypass the “Comments to the Author” section, enter your conflict of interest statement in the “Confidential to Editor” section, and submit your "Accept" recommendation.

Reviewer #2: All comments have been addressed

2. Is the manuscript technically sound, and do the data support the conclusions?

Reviewer #2: Yes

3. Has the statistical analysis been performed appropriately and rigorously? 

Reviewer #2: Yes

4. Have the authors made all data underlying the findings in their manuscript fully available?

Reviewer #2: Yes

5. Is the manuscript presented in an intelligible fashion and written in standard English?

Reviewer #2: Yes

6. Review Comments to the Author

Reviewer #2: The research, using artificial neural network to develop what I called Punch mechanism has been improved and i have no further objection

7. PLOS authors have the option to publish the peer review history of their article (what does this mean?). If published, this will include your full peer review and any attached files.

Reviewer #2: No

---

## [Editor Report · Acceptance letter]

5 Nov 2021

PONE-D-21-15481R1 

Using an artificial neural network to develop an optimal model of straight punch in boxing and training in punch techniques based on this model and real-time feedback 

Dear Dr. Khasanshin:

I'm pleased to inform you that your manuscript has been deemed suitable for publication in PLOS ONE. Congratulations! Your manuscript is now with our production department. 

Kind regards, 

on behalf of

Dr. Bijan Najafi 

Academic Editor

PLOS ONE